# *Lactobacillus reuteri* MJM60668 Prevent Progression of Non-Alcoholic Fatty Liver Disease through Anti-Adipogenesis and Anti-Inflammatory Pathway

**DOI:** 10.3390/microorganisms10112203

**Published:** 2022-11-07

**Authors:** Pia Werlinger, Huong Thi Nguyen, Mingkun Gu, Joo-Hyung Cho, Jinhua Cheng, Joo-Won Suh

**Affiliations:** 1Interdisciplinary Program of Biomodulation, Myongji University, Yongin 17058, Korea; 2Myongji Bioefficacy Research Center, Myongji University, Yongin 17058, Korea

**Keywords:** *Lactobacillus reuteri* MJM60668, NAFLD, anti-adipogenesis, probiotics, microbiota

## Abstract

Non-alcoholic fatty liver disease (NALFD) is a disease characterized by liver steatosis. The liver is a key organ involved in the metabolism of fat, protein, and carbohydrate, enzyme activation, and storage of glycogen, which is closely related to the intestine by the bidirectional relation of the gut-liver axis. Abnormal intestinal microbiota composition can affect energy metabolism and lipogenesis. In this experiment, we investigated the beneficial effect of *Lactobacillus reuteri* MJM60668 on lipid metabolism and lipogenesis. C57BL/6 mice were fed a high-fat diet (HFD) and orally administrated with MJM60668. Our results showed that mice treated with MJM60668 significantly decreased liver weight and liver/body weight ratio, without affecting food intake. Serum levels of ALT, AST, TG, TCHO, and IL-1β in mice fed with MJM60668 were decreased compared to the HFD group. Investigation of gene and protein expression on the lipogenesis and lipid metabolism showed that the expression of ACC, FAS, and SREBP was decreased, and PPARα and CPT was increased. Furthermore, an increase of adiponectin in serum was shown in our experiment. Moreover, serum IL-1β level was also significantly decreased in the treated mice. These results suggested that MJM60668 can strongly inhibit lipogenesis, enhance fatty acid oxidation, and suppress inflammation. Additionally, supplementation of MJM60668 increased the proportion of *Akkermansiaceae* and *Lachnospiracea*, confirming a potential improvement of gut microbiota, which is related to mucus barrier and decrease of triglycerides levels.

## 1. Introduction

Non-alcoholic fatty liver disease (NAFLD), a worldwide disease of interest with a dramatic escalation, is affecting around one-third of the population [1]. It has been reported that NAFLD affects one in four adults with a global prevalence of 25.24% [2]. Due to its increasing prevalence, NALFD is becoming the most common chronic liver disease these days. It has been predicted to be the most frequent indicator for liver transplantation by 2023 [3]. NAFLD starts with a simple hepatic steatosis. It continues with necroinflammation through different stages of fibrosis, finally leading to the development of nonalcoholic steatohepatitis (NASH) characterized by liver cirrhosis as a chronic stage [4].

NAFLD is characterized by fat accumulation and inflammation in the hepatocytes, increasing levels of TNF-α stimulating chronic liver inflammation, progressing to steatosis and fibrosis of the liver parenchyma, leading to cirrhosis, hepatocellular carcinoma, and liver failure as an end-stage [5,6]. Adipogenesis refers to the process of developing and accumulating adipocytes in the body with the function of control metabolism by secretion of leptin and adiponectin [7]. In the process of adipogenesis, adipocyte experiences a triglycerides accumulation leading to an expansion of adipocyte depots and accumulation of lipid droplets [8,9] inducing an increase of ROS levels leading to an inflammation process in organs [10]. Lipid droplets are a key component involved in hepatic metabolism, under physiological conditions lipids droplets are stored in the liver activating the proliferation of HSD and enhancing the resistance of apoptosis of hepatocytes via activation of the PI-3-kinase pathway. This process will activate a fibrogenic process, responsible for steatosis development. For this reason, droplet lipid accumulation in the liver is the hallmark of NALFD [11].

Intestinal microbiota are a group of millions of diverse microorganisms that consistently interact with the host [12] with the majority of them residing in distal segments of the gastrointestinal tract [13]. Numerous microorganisms are classified as probiotics, which are living microorganisms with beneficial effects on the gastrointestinal tract of the host [14]. Bacteria belonging to *Lactobacillus* and *Bifidobacterium* genera are the most common probiotics. They show bile acid resistance associated with the activation of glycolysis [15]. Therefore, intestinal microbiota plays a key role in metabolism, and regulation of intestinal microbiota might be a promising strategy for NAFLD treatment.

*Lactobacillus* spp. is a large group of Gram-positive bacteria with an important role in food fermentation. They can be found in the gastrointestinal tract of mammals [16,17]. Studies have shown that *Lactobacillus* can help prevent and treat enteric infections, antibiotics diarrhea, enterocolitis, colorectal cancer, and inflammatory disease by improving gastrointestinal dysbiosis [18]. *Lactobacillus reuteri* is a heterofermentative species that can grow in oxygen-limited atmospheres and easily colonize the gastrointestinal tract and confer excellent probiotic properties, surviving in different pH levels and secreting antimicrobial intermediaries [19,20]. Different studies have shown that the administration of *L. reuteri* can promote health in humans [21]. A previous study revealed *Lactobacillus reuteri* can reduce fat accumulation, steatosis [22], and fibrosis [23] in high-fat-diet-treated mice. However, the mechanism was not fully understood. Also, *L. reuteri* strain has shown a positive effect in LPS-induced intestinal tight junction protein destruction, decreasing intestinal and inhibiting intestinal and hepatic inflammatory signals in piglets [24]. Additionally, it has shown an effect as a positive regulator of gut microbiota and improves the metabolic system by increasing short-chain fatty acids (SCFAs) [25].

Therefore, in the present study, we studied the effect of *L. reuteri* MJM60668 on NAFLD in a high-fat induced mice model because this strain showed strong inhibitory activity to lipid accumulation in HepG2 hepatic cells, as well as good probiotic characteristics. In addition, the underlying mechanism was also investigated through anti-adipogenesis and anti-inflammatory pathways.

## 2. Materials and Methods

### 2.1. Strain Identification by 16S rDNA Sequence and Phylogenetic Analysis of MJM60668

Strain MJM60668 was isolated from fecal samples of baby infants and stored at the lactobacillus library in our laboratory. During the screening of the useful Lactobacillus project in our laboratory, strain MJM60668 showed good probiotic characteristics and thus was selected for further study. To identify lactic acid bacteria (LAB), 16S rDNA sequencing was performed to identify species. Genomic DNA was isolated from MJM60668 using an Exgene cell SV DNA isolation kit (GneAll, Seoul, Korea). The 16S rDNA sequence was amplified with universal primers 27F and 1492R and sequenced. Obtained sequences were blasted against the Ezbiocloud database (ChunLab Inc., Seoul, Korea). Finally, 16S rDNA sequences of closely related strains were aligned and a phylogenetic tree was constructed by the neighbor-joining method using MEGA X software [26].

### 2.2. Cell Culture and Treatment

To assess the anti-lipidemic activity, HepG2 cells obtained from Korean Cell Line Bank (KCLB) were cultured in DMEM media (Sigma-Aldrich, St. Louis, MO, USA) supplemented with 10% FBS and 1% antibiotics (streptomycin/penicillin solution, Gibco, CA, USA) at 37 °C in humidifier chamber containing CO_2_. Cells were seeded and treated with samples for 6 h following previously published methods [27,28], with some modifications. Briefly, HepG2 cells were seeded at a density of 10^6^ cells/well in 2 mL of medium, incubated for 24 h, and then starved for 24 h in an FBS-free culture medium. HepG2 cells were then treated with or without fatty solution (1 mM Oleic acid, 7.5 µg/mL cholesterol) containing probiotics via the insert Transwell membrane (SPL, Pochon, Korea), which was seeded with 10^8^ to 10^9^ CFU/mL MJM60668. Simvastatin (1 µM) was used as a positive control.

### 2.3. Cell Viability Assay (MTT)

The MTT ((3-(4,5-dimethylthiazol-2-yl)-2,5 diphenyl tetrazolium bromide) assay was used to determine the toxicity of MJM60668 to HepG2 cells. After 6-h treatment with samples, transwell membranes were removed and 20 µL MTT (5 mg/mL) was added to each well followed by incubation for an additional 4 h. Finally, the medium was removed and then 200 µL DMSO was added to each well to dissolve formazan crystals, the metabolite of MTT. The absorbance of each well was measured at a wavelength of 570 nm using a microplate reader (TECAN Spectrofluor Plus, Maennedorf, Switzerland). The viability of the non-treatment group was set to 100%.

### 2.4. Oil Red O Staining

Intracellular lipid accumulation was measured by Oil Red O staining. Briefly, 0.2 g of Oil red O powder (Sigma-Aldrich, St. Louis, MO, USA) was dissolved in 40 mL 2-propanol (0.5% *w/v*) to obtain a stock solution. The working solution was obtained by diluting the stock solution at 2:3 with distilled water. It was filtered with a 0.2 µm syringe filter before use.

After 6 h of treatment, HepG2 cells were washed with PBS and fixes twice with 10% formaldehyde (5 min for the first time and 1 h for the second time) before staining with the Oil red O working solution. After 30 min, cells were washed twice with distilled water and photographed under a microscope. Finally, the dye retained in HepG2 cells was eluted by adding 100% isopropanol and quantified at a wavelength of 510 nm with a microplate reader (TECAN Spectrofluor Plus, Maennedorf, Switzerland).

### 2.5. Safety Test

#### 2.5.1. D-Lactate Production

D-lactate is generated by D-lactate dehydrogenase under hypoxic or anaerobic conditions and is only produced in low quantity in animals under normal conditions because D-lactate is a specific indicator of bacterial fermentation. D-lactate is also produced in the human colon under normal conditions by the fermentative process of lactic acid produced by Gut microorganisms, however, D-lactate increasing in the gastrointestinal tract can develop D-lactic acidosis which is responsible for triggering multiple problems in the liver and kidneys making these organs unable to remove acid from the body [29].

D-Lactate production of MJM60668 was determined using a D-lactate enzyme test kit (Megazyme, Ireland). LGG was used as a control group. Then, 0.01 mL of LAB supernatant culture was mixed with 0.15 H_2_O, 0.05 mL supplied buffer (pH 10), 0.01 mL of NAD+ solution, and 0.002 mL of D-glutamate-pyruvate transaminase (D-GPT) and incubated at room temperature for 3 min. After, the absorbance of the solution was measured at 340 nm wavelength (TECAN Spectrofluor Plus, Maennedorf, Switzerland). Next, 0.002 mL of 2000 U/mL D-lactate dehydrogenase (D-LDH) was added to the above reaction mixture, and absorbance was measured for 5 min until the D-LDH reaction stopped. Finally, the solution absorbance was measured at 340 nm wavelength (TECAN Spectrofluor Plus, Maennedorf, Switzerland). Concentrations of D-lactate were calculated according to the manufacturer’s equation.

#### 2.5.2. Bile Salts Deconjugation

To this, MJM60668 was placed on MRS agar plates supplemented with 0.5% (*w/v*) sodium salt of Sodium taurodeoxycholate hydrate. The plates were incubated at 37 °C for 2 and 5 days. After, the diameters of the deconjugated bile acid precipitation zones (opaque halos) were measured. LAB strains placed on MRS agar medium plates without supplementation of the conjugated bile acids were used as a control and LGG was a positive control group.

#### 2.5.3. Antimicrobial Susceptibility Test (MIC)

MJM60668 and LGG were evaluated for susceptibly to nine antibiotics used to treat enterococcal infections (ampicillin, vancomycin, gentamycin, kanamycin, streptomycin, erythromycin, clindamycin, tetracycline, and chloramphenicol) recommended by European Food Safety Authority (EFSA, 2012) [30]. The minimum inhibitory concentration (MIC) with different antibiotics of LAB strains was determined using the microdilution method [31]. The MIC values of MJM60668 were compared with the EFSA cutoff values for *Lactobacillus reuteri*.

#### 2.5.4. Hemolytic Activity

The test method was referred to the American Society for Microbiology guidelines. Hemolytic activity of the MJM60668 and LGG was tested by streaking to 5% sheep blood in tryptic soy agar (Hardy Diagnostics, A10) and incubated at 37 °C for 24 h. Hemolytic activity was determined by the measurement of the photochromic properties around the colonies under transmitted light.

#### 2.5.5. Biogenic Amine Production Test

Decarboxylase medium with or without supplemented 1% amino acid (L-histidine, L-tyrosine, L-phenylalanine, Arginine, Tryptophan, and L-ornithine) was used to determine biogenic amine production of MJM60668 following previous study [32]. The components of decarboxylase medium include 0.5% Tryptone, 0.5% yeast extract, 0.5% beef extract, 0.25% NaCl, 0.05% glucose, 0.1% tween 80, 0.02% Magnesium sulfate, 0.005% Manganese sulfate, 0.004% Iron (II) sulfate, 0.2% Ammonium citrate, 0.001% Thiamine, 0.2% Dipotassium hydrogen phosphate, 0.01% calcium carbonate, 0.005% pyridoxal 5-phosphate, 0.006% bromocresol purple, 2% agar, pH 5.5. The medium without added amino acids was used as the control.

#### 2.5.6. Mucin Degradation Activity

Mucin degradation activity was tested by using 0.3% porcine gastric mucin in an agarose medium with or without glucose supplementation. Then, 10 μL of MJM60668 was added to the medium and cultured at 37 °C, for 72 h. After that, the medium was dyed with 0.1% amido black in acetic acid (3.5 M) for half-hour and washed with acetic acid (1.2 M) until the clearance zone was detected in the positive control. The determination of mucin degradation activity was conducted via measurement clearance zone around the colony.

#### 2.5.7. Antimicrobial Assay

MJM60668 was tested with eight pathogenic strains including *Salmonella gallirarum* KCTC 2931, *Escherichia coli* K99, *Escherichia coli* O1 KCTC 2441, *Escherichia coli* 0138, *Escherichia coli* ATCC25922, *Salmonella chloreraesuis* KCTC 2932, *Salmonella typhi* KCTC 2514, and *Pseudomonas aeruginosa* KCCM 11802 using the agar well diffusion method. Then, 100 μL cell-free culture medium of MJM60668 or LGG was added to 8 mm wells of agar plates containing individual pathogenic strains and incubated at 37 °C for 1 day. Finally, antimicrobial activity was determined by measuring the diameters of inhibition zones except 8 mm of well. The assay was conducted in triplicates.

#### 2.5.8. Oro-Gastrointestinal Transit Assay

MJM60668 and LGG (positive control) with a density of 10^9^ CFU/mL were initially subjected to oral stress solution (0.62% sodium chloride, 0.22% potassium chloride, 0.022% calcium chloride, 0.12% sodium bicarbonate and 0.015% lysozyme) at 37 °C for 10 min. After centrifugation at 1800× *g* for 5 min, the supernatant was removed and the cells were resuspended to gastric stress solution (0.3 g/0.1 L pepsin, adjust pH to 3) for a half-hour, then treated with gastric stress solution at pH 2 for a half-hour. After centrifugation, the gastric solution was removed and the intestinal solution (0.5% sodium chloride, 0.06% potassium chloride, 0.025% calcium chloride, 0.1% pancreatin, 0.3% bile bovine, pH 7) was added for 2 h. LAB, which was treated only with PBS for the entire assay duration, was used as the control. Cells after each step were diluted, plated on MRS agar plates, and incubated for 48 h. The cell viability of LAB was calculated based on the number of colonies grown on MRS agar plates.

#### 2.5.9. Cell Adhesion Assay

To assess the effectiveness of MJM60668 to adhere to the gastrointestinal tract, the cell adhesion assay was performed.

HT-29 cells were purchased from Korean Cell Line Bank (KCLB) and cultured RPMI 1640 (HyClone Laboratories, Inc., Logan, UT, USA supplemented with 10% FBS (Gibco, Waltham, MA, USA), 100 U/mL penicillin and 100 mg/mL of streptomycin (Gibco, Waltham, MA, USA), was incubated at 37 °C on 5% CO_2_. HT-29 cells were seeded at concentrations of 2 × 10^5^ cells/well in 24 well plates and incubated for 24 h to get polarized monolayer and 80% confluence.

HT-29 monolayers were washed three times with PBS, the medium was replaced with antibiotic-free RPMI 1640. Each well was inoculated with bacteria (final concentrations 1 × 10^8^ CFU/mL) and incubated for 2 h at 37 °C in a 5% CO_2_ incubator. After incubation, monolayers were washed three times with PBS and treated with 1000 µL of Trypsin (Sigma-Aldrich, St. Louis, MO, USA) for 5 min. Then, 1 mL aliquot of homogenate was serially diluted and plated on MRS agar and incubated at 37 °C for 48 h. The percentage of bacteria adhered to the plate was calculated by dividing the remaining bacteria grown on MRS agar by the initial inoculation bacteria. All experiments were performed in triplicate.

### 2.6. Animal Assay

For this study, 7-week-old C57BL/6 male mice bought from KOATECH (Gyeonggi, Korea) were randomly assigned into cages with four mice in each of them in a room with 12-h light cycle (lights on at 6:00 am and off at 6:00 pm) at 22 ± 2 °C and 55 ± 5 of humidity and with water and feed freely supplied for 1-week prior experiment. 

All protocols for animal experiments were approved by the Institutional Animal Care and Committee (IACUC) of Myongji University (MJIACUC-2021003) and thus conducted in accordance with the NIH guide for the care and Use of Laboratory Animals. 

#### 2.6.1. Animal Monitoring and Treatment

After acclimatization for 1 week, mice were divided into five groups (n = 12 mice per each group): Control, HFD, Silymarin, MJM60668 (10^8^), and MJM60668 (10^9^) groups. Control group mice were fed a normal rodent diet from Raon Bio (Gyeonggi, Korea). Groups HFD, Silymarin, MJM60668 (10^8^), and MJM60668 (10^9^) were fed a high-fat diet (HFD) composed of 45% fat, 35% carbohydrates, and 20% proteins (Raon Bio, Gyeonggi, Korea) to increase adipocyte infiltration in the liver and induce steatohepatitis (NASH) [33].

Animals in control and HFD groups were treated orally with saline solution. Group Silymarin mice were treated with silymarin (50 mg/kg) (Sigma-Aldrich, St. Louis, MO, USA). Group MJM60668 (10^8^) mice were treated with *L. reuteri* MJM60668 at 1 × 10^8^ CFU/mice/day. Group MJM60668 (10^9^) mice were treated with *L. reuteri* MJM60668 at 1 × 10^9^ CFU/mice/day. All treatments were administrated orally once daily for 12 weeks. LABs were cultured freshly daily, washed, and resuspended in saline (0.85% NaCl) before use.

Mice were monitored daily. Parameters such as body weight, activity, appearance, and posture were recorded twice a week to evaluate the comfort of animals and give them scores from 1 to 5. If any animal had a score higher than 3 (losing weight higher than 20% compared with other animals in the same cage, hair problems, or showing abnormal activities), mice were taken out and immediately euthanized. After 12 weeks of treatment, mice were euthanized by cervical dislocation after anesthesia induction with 3% isoflurane. Organs were then taken for analysis.

#### 2.6.2. Serum Biochemical Analysis

Blood samples were collected from the heart at 4 and 12 weeks of treatment. Collected samples were immediately moved onto ice. After 1 h, samples were centrifuged at 2000× *g* for 15 min at 4 °C. The serum was separated and stored at −80 °C. Separated plasma samples were analyzed with FUJIFILM DRI-CHEM NX500i to measure the contents of Aspartate transferase (AST), Alanine Aminotransferase (ALT), Creatinine (CREA), Uric Acid (UA), Blood Urea Nitrogen (BUN), Total Triglycerides (TG), Gamma-glutamyl Transferase (GGT), Albumin (ALB), Total Cholesterol (TCHO), and High-Density Cholesterol (HDL-C).

#### 2.6.3. Liver, Kidney, Intestine Sectioning, and Histopathological Assessment

After mice were euthanized, the liver, kidney, and intestine were immediately removed and rinsed. Liver and kidney weights were recorded. The liver was sectioned into three parts: median and left lobe, right lobe, and caudate lobe.

The caudate lobe and right kidney were fixed in 4% paraformaldehyde (Sigma-Aldrich, St. Louis, MO, USA) for 24 h. They were then paraffin-embedded, sliced into 5 µm thick sections, and stained with hemoxylin-eosin (H&E). Histological changes were examined at 200× magnification using a computer image analysis system (CaseViewer by 3DHISTECH, Budapest, Hungary). Each image was analyzed and graded according to percentages of inflammation and vacuolation of hepatocytes to three levels: mild (0–25%), moderate (26–50%), and severe (>51%) as described before [34,35].

#### 2.6.4. Western Blot Analysis

Frozen median and left lateral lobe of liver samples were ground with liquid nitrogen to powder. RIPA lysis buffer supplemented with 1% protease and phosphatase inhibitors was added to each sample and incubated on ice for 30 min. After that, samples were centrifuged at 14,000 rpm for 10 min at 4 °C. The supernatant was collected and transferred to a new tube. Total protein was measured with a BCA Protein Assay Kit (Thermo Fisher Scientific, Rockford, IL, USA). Proteins were boiled and loaded (40 µg/lane) alongside a molecular weight marker (Thermo Fisher Scientific, Rockford, IL, USA) on a 10% SDS-polyacrylamide gel (SDS-PAGE). After the gel was run, proteins were transferred to a polyvinylidene fluoride (PVDF) membrane. After the membrane was blocked with 3% BSA (Bovine Serum Albumin) in TBS (Tris-buffered saline) for 1 h at 4 °C, the membrane was incubated with primary antibodies against FAS (1:200, Abcam, Cambridge, MA, USA), SREBP (1:1000, Abcam, MA, USA), PPARα (1:1000, Abcam, MA, USA), or β-actin (1:10,000, Abcam, MA, USA). After washing three times with TBST buffer (10 mM Tris, 150 mm NaCl, and 0.1% Tween-20), the membrane was further incubated with HRP-conjugated goat anti-rabbit IgG secondary antibody (1:10,000, Abcam, MA, USA) for 90 min at 4 °C. After washing the membrane three times with TBST buffer, immunoreactive bands were enhanced using chemiluminescence reagents. Membrane photos were then captured. Expression levels of target proteins were compared to β-actin protein levels using ImageJ software (U.S. National Institutes of Health, Bethesda, MD, USA) to obtain quantitative protein expression. All experiments were performed in triplicate.

#### 2.6.5. RNA Extraction, cDNA Synthesis, and Quantitative Real-Time PCR (qRT-PCR)

Frozen samples of the right lateral lobe of the liver were ground with liquid nitrogen to powder. Total RNA was isolated with a Takara MiniBEST Universal RNA extraction kit (Takara, Shiga, Japan) according to the manufacturer’s instructions. After confirming the integrity of the RNA sample by agarose gel electrophoresis, RNA concentration and purity were determined with an ND-1000 spectrophotometer. For cDNA synthesis, gDNA was removed from the RNA with a gDNA eraser (Takara, Dalian, China). After cDNA was synthetized using a Takara PrimeScript RT reagent kit following the manufacturer’s instruction, 1 µg of cDNA was mixed with 10 µL of SYBR Premix Ex Taq (Takara, Dalian, China) and 0.5 µM of each primer of a primer pair in 20 µL Roche LightCycler^®^ 96 system. Primers used in this study are listed in Table 1. The ratio of each gene was compared with β-actin to standardize the ratio for each control to the unit value. Relative mRNA levels of genes including fatty acid synthase (FAS), Acetyl-CoA carboxylase (ACC), peroxisome proliferator-activated receptor alpha (PPARα), carnitine palmitoyltransferase 1a (CPT1A), and interleukin 6 (IL-6) were determined and normalized to the expression of β-actin using the 2 ^−ΔΔCT^ method [36].

#### 2.6.6. Fecal Sample Analysis

The intestine was immediately dissected. Feces were collected into tubes and stored at −80 °C before use. For metagenome analysis, DNA was extracted with an Exgene^TM^ Stool DNA mini kit (GeneAll, Seoul, Korea) according to the manufacturer’s protocol. The V3-V4 region of the bacterial 16S rRNA gene was amplified using barcoded universal primers 341F and 805R [37]. Microbiome profiling was conducted with a 16S-based Microbial Taxonomic Profiling platform of EzBioCloud Apps (Accugene, Gyeonggi, Korea).

### 2.7. Statistical Analysis 

All data are shown as mean ± SD. The difference between the control and experimental groups was subjected to the one-way analysis of variance (ANOVA) or two-sample *t*-test to determine statistical significance. Differences between groups were considered statistically significant when the *p*-value was equal to or less than 0.05.

## 3. Results

### 3.1. Phylogenic Analysis of MJM60668

To accurately determine the species of bacteria isolated, 16 rDNA gene was sequenced. Phylogenetic analysis was then performed. A blast search of the Ezbiocloud database showed that the 16S rDNA sequence of the isolate was identical to *Limosilactobacillus reuteri* species, sharing 99% similarity. This indicates that MJM60668 is closely related to *L. reuteri* (Figure 1).

### 3.2. Effect of MJM60668 on Viability of HepG2 Cells

MTT assay was used to assess the cytotoxicity of MJM60668 to hepatocytes. As shown in Figure 2a, there was no significant difference in the viability of HepG2 cells among treatment groups. Treatment with *L. reuteri* MJM60668 at both concentrations (1 × 10^8^ CFU/mL and 1 × 10^9^ CFU/mL) had no cytotoxicity to HepG2 cells. On the other hand, it slightly improved the survival rate of HepG2 cells compared to OA-C treatment. This result suggests that MJM60668 is safe for hepatocytes (Figure 2a).

### 3.3. Anti-Lipogenic Effect of MJM60668 on HepG2 Cells

HepG2 cells were treated with MJM60668 at two doses (1 × 10^8^ and 1 × 10^9^ CFU/mL) in a medium containing fatty solution (Oleic acid and Cholesterol) for 6 h to measure lipid accumulation level. At 1 × 10^8^ and 1 × 10^9^ CFU/mL, MJM60668 reduced lipid accumulation by 50.5% and 81.29%, respectively, compared to the negative control (OA-C, 173.76%) (Figure 2b). Moreover, the inhibitory effect of MJM60668 on lipid accumulation was stronger than that of the positive control (simvastatin) which decreased lipid accumulation by 36.89% compared to the OA-C (Figure 2b).

### 3.4. Safety Assessment of MJM60668 

#### 3.4.1. D-Lactic Production, Bile Salt Deconjugation, Hemolytic Activity, Mucin Degradation Activity, and Antibiotic Susceptibility

MJM60668 showed non-hemolytic activity on blood agar (Table 2). Moreover, MJM60668 did not exhibit D-lactate production, bile salt deconjugation, or bioamine production. In addition, similar to LGG, MJM60668 was sensitive to almost all antibiotics tested except for gentamycin, streptomycin, and kanamycin.

#### 3.4.2. Antibacterial Activity of MJM60668

The antimicrobial activity of MJM60668 is shown in Table 3. MJM60668 and LGG had similar antimicrobial activities against all pathogens tested. Both MJM60668 and LGG strongly inhibited seven pathogens (*Escherichia coli* K99, *Salmonella gallirarum* KCTC 2931, *Escherichia coli* O1 KCTC 2441, *Escherichia coli* ATCC25922, *Salmonella chloreraesuis* KCTC 2932, *Salmonella typhi* KCTC 2514, and *Pseudomonas aeruginosa* KCCM 11802). MJM60668 and LGG also showed moderate inhibitory activities against *Esherichia coli* O138 (Table 3).

#### 3.4.3. Adherence of MJM60668 to HT-29 Cells

The adherence of MJM60668 and LGG to HT-29 cells was assessed by the percentage of LAB number binding to a monolayer of HT-29 cells after 2 h of incubation and three times of washing compared to the initial LAB added. The adherent ability was 5.2% for MJM60668. It was only 3.19% for LGG (Table 2).

#### 3.4.4. OGI Transit Assay

Survival of MJM60668 under oral, gastric, and intestinal stresses was tested with an oro-gastro-intestinal (OGI) transit assay. MJM60668 and LGG were not affected by oral stress (*p* = 0.31 and *p* = 0.44, respectively). However, log CFU units of both MJM60668 and LGG were significantly decreased after gastric stress by 0.23 and 0.38, respectively, compared with the initial CFU. MJM60668 and LGG showed significant decreases of 0.52 and 1.33 log CFU units (both *p* < 0.01), respectively, at the end of the OGT assay compared to the initial CFU (Table 4).

### 3.5. Animal Study

#### 3.5.1. Effects of MJM60668 on Body Weight, Food Intake, Various Tissue Weights on HFD-Induced NAFLD Model in Mice

As shown in Figure 3b, the HFD group of mice showed slight differences in body weight starting 3 weeks after treatment compared with the control. Nevertheless, after 12 weeks of treatment, the body weights of the HFD group of mice were significantly increased (42.23 ± 2.34 g) compared with those of the control group mice (32.34 ± 2.63 g). After administration of silymarin or MJM60668, the increase in body weight was significantly decreased, reaching levels similar to those of the control group. Body weights of mice in groups treated with silymarin, MJM60668 (10^8^), and MJM60668 (10^9^) after 12 weeks of treatment were 34.23 ± 1.92, 35.96 ± 3.88, and 35.88 ± 3.67 g, respectively.

Additionally, our results showed an increase in food consumption of the control group compared to HFD feed groups (Figure 3c). This can be explained by the fact that HFD mice can consume extra calories due to the high amount of calories provided by the food without increasing the amount of food consumed [38,39].

After sacrifice, morphologies of mice livers were compared between groups. Results showed that liver sizes of HFD-treated mice were increased. Additionally, the livers of mice in the HFD group showed a whitish tone compared to those in other groups (Figure 3e). These results suggest the effectiveness of HFD in inducing NAFLD in mice after 12 weeks of administration. In the case of the mice fed with the HFD diet but treated with silymarin or MJM60668 strain at different concentrations, they showed slightly increased body weights compared to mice of the control group. However, their livers only showed a slight difference in coloration compared with the livers of the control group, whereas they showed no significant difference in liver weight.

An increase of visceral adipose tissue can increase free fatty acid secretion into portal circulation accompanied by compensatory hyperinsulinemia which can stimulate fatty acid synthesis and inhibit catabolism of fatty acid in hepatocytes. When the amount of fatty acids exceeds the capacity of beta-oxidation, acetyl CoA drained by triglyceride synthesis will accumulate, leading to liver steatosis [40]. After sacrifice, body weight, liver weight, and epididymal fat were measured to confirm the effectiveness of MJM60668 in decreasing fat accumulation in the liver and the relation of MJM60668 treatment with body weight and epididymal fat decreases of treated mice. Our results showed significantly decreased body weights in Silymarin, MJM60668 (10^8^), and MJM60668 (10^9^) groups (33.77 ± 2.44 g (*p* ≤ 0.0001), 33.98 ± 2.05 g (*p* ≤ 0.0001), and 33.84 ± 2.76 g (*p* ≤ 0.0001), respectively) compared to the HFD group (42.93 ± 2.36 g) (Figure 4a). Additionally, liver weights of treated mice of MJM60668 (10^8^) and MJM60668 (10^9^) groups were significantly decreased to 1.22 ± 0.17 g (*p* ≤ 0.0001), and 1.17 ± 0.2 g (*p* ≤ 0.0001), respectively, compared to those of the HFD group (1.76 ± 0.3 g) (Figure 4b). After sacrifice, epididymal weight was measured for each group. Results showed that epididymal weights of MJM60668 (10^8^) and MJM60668 (10^9^) groups (0.3 ± 0.06 g and 0.2 ± 0.04 g, respectively) were slightly decreased compared with those of the HFD group (0.33 ± 0.06 g) (Figure 4c).

Finally, after measuring the body weight and liver weight of each animal, their ratio was calculated. As shown in Figure 4d, body weight to liver weight ratios were significantly decreased in Silymarin (3.76 ± 0.2 g) (*p* = 0.0026), MJM60668(10^8^) (3.68 ± 0.32 g) (*p* = 0.0009), and MJM60668(10^9^) (3.53 ± 0.49 g) (*p* = 0.0001) groups, compared with that of the HFD group (4.44 ± 0.24 g). This result suggests that MJM60668 is a potential strain that can be used to decrease fatty acid accumulation in the liver, thus improving weight loss.

#### 3.5.2. Effects of MJM60668 on Serum and Hepatic Lipid Profile in HFD-Induce NAFLD Mice Model

Previous studies have indicated that ALT and AST are the main markers of liver injury in NALFD [41]. ALT and AST are mainly located in hepatocellular cytosol and mitochondria, respectively. Together they can indicate liver conditions. They significantly increase under body weight gain and steatosis presence in humans [41,42,43,44]. In the present study, the HFD group showed higher serum levels of ALT and AST than the control group (HFD: 197.11 ± 118.85 ng/mL and 41.85 ± 10.86 ng/mL, respectively; Control: 30.6 ± 5.09 ng/mL and 56.6 ± 15 ng/mL, respectively). Administration of MJM60668 (10^8^) and MJM60668 (10^9^) significantly reduced the increased levels of AST (to 66.88 ± 23.22 µmol/L and 50.66 ± 14.62 µmol/L, respectively) and ALT (to 17.62 ± 2.97 µmol/L and 16 ± 1.41 µmol/L, respectively) seen in the HFD group (Figure 5a,b).

To determine whether MJM60668 could suppress the formation of steatosis, serum levels of triglycerides (TG) and total cholesterol (TCHO) were measured. Our results showed significantly increased TG and TCHO levels in the HFD group (156.53 ± 45.39 µmol/L and 200.8 ± 15.78 µmol/L, respectively) than in the control group (96.46 ± 33.32 µmol/L and 121.8 ± 16.68 µmol/L, respectively). Such increased hepatic TG and TCHO levels were suppressed by the administration of MJM60668 (10^8^) (to 96.4 ± 29.28 µmol/L and 178.5 ± 23.52 µmol/L, respectively) and MJM60668 (10^9^) (to 79.41 ± 23.31 µmol/L and 163.62 ± 23.21 µmol/L, respectively) (Figure 5c,d).

As shown in Figure 5e, HDL-C levels were increased in the HFD group (110 ± 0 µmol/L) than in the control group (100.58 ± 8.05 µmol/L). However, HDL-C levels in MJM60668 (10^8^) and HFD groups showed no significant difference (110 ± 0 and 110 ± 0 µmol/L, respectively). Nevertheless, the MJM60668 (10^9^) group showed a slight decrease in HDL-C level (109.5 ± 1.08 µmol/L) compared with the HFD group, suggesting that MJM60668 could be potentially used as a probiotic to decrease HDL levels in the body in a dose-dependent manner

#### 3.5.3. MJM60668 Can Improve Liver Regeneration under Steatosis Based on Histological Analysis

NAFLD is characterized by an increase of intracytoplasmic lipids and ballooned hepatocytes (a special form of hepatocyte degeneration) as the principal histological findings in steatohepatitis [45,46]. In our experiments, ballooned hepatocytes (Figure 6b) were with a score ranging from 0 (normal) to 3 (severe) [35,47,48]. To examine the grade of steatosis in the liver, H&E staining was performed for four mice per group to measure the steatosis grade per sample. As shown in Figure 6a, ballooned hepatocytes in the HFD showed a significantly higher grade (grade 3) of steatosis than the control group (grade 0). Administration of MJM60668 significantly decreased the steatosis grade induced by HFD (steatosis grade: MJM60668 (10^8^) group, 2 ± 0.8; and MJM60668 (10^9^) group, 1.5 ± 0.57; HFD group, 3 ± 0). Thus, MJM60668 could dose-dependently influence liver regeneration and reverse steatosis induced by high-fat diet.

#### 3.5.4. Effects of MJM60668 on ACC, Fas, PPARα, CPT1A, and IL-6 Gene Expression in HFD Group of Mice

Expression levels of ACC, Fas, PPARα, CPT1A, and IL-6 mRNA levels in the liver are shown in Figure 7. Expression levels of genes associated with fatty acid syntheses such as ACC and Fas mRNAs and inflammation-associated genes such as IL-6 mRNA in the liver were significantly increased in the HFD group (2.35 ± 0.46, 2.39 ± 0.23, 2.09 ± 0.43, respectively) compared with those in the control group (1 ± 0.12, 1 ± 0.16, and 1 ± 0.13, respectively). Additionally, following treatment with MJM60668, ACC mRNA, Fas mRNA, and IL-6 mRNA expression levels were significantly decreased (0.97 ± 0.6, 1.6 ± 0.15, and 1.8 ± 0.39, respectively) compared with those in the HFD group (Figure 7a,b,e). Nevertheless, expression levels of genes involved in lipid metabolism and fatty acid oxidation (PPARα mRNA and CPT1A mRNA) were significantly reduced in the HFD group (0.37 ± 0.05 and 0.42 ± 0.03, respectively) compared with those in the control group (1 ± 0, 1 ± 0, respectively). Administration of MJM60668 significantly increased mRNA expression levels of PPARα (3.1 ± 0.4) and CPT1A (2.35 ± 0.5) compared to the control (1 ± 0) (Figure 7c,d).

#### 3.5.5. Effects of MJM60668 on Proteins Involved in Fatty Acid Synthesis and Lipid Metabolism

As shown in Figure 8a,b, protein expression levels of FAS and SREBP in liver tissues were significantly increased in the HFD group (3.65 ± 0.11, and 1.21 ± 0.24) compared to the control (1 ± 0.04, and 0.83 ± 0.03, respectively). Nevertheless, HFD-fed mice after treatment with MJM60668 showed significantly decreased protein expression levels of FAS and SREBP (1.706 ± 0.07 and 0.67 ± 0.07, respectively) compared with the control group. SREBP is a gene involved in lipogenesis and glycolysis [49]. FAS is involved in the apoptosis of hepatocytes and steatosis progression [50]. Our results suggest that MJM60668 can strongly and significantly inhibit lipogenesis and decrease the conversion of carbohydrates into fatty acids.

PPARα has a key role in lipid metabolism and glucose homeostasis under the catabolism of fatty acids [51] by regulating the expression of genes involved in fatty acid oxidation [52]. PPARα protein expression in liver tissues was significantly decreased in the HFD group (0.72 ± 0.05) compared with that in the control group (1 ± 0.01). Nevertheless, administration of MJM60668 (HFD + MJM60668) significantly increased PPARα protein expression (1.25 ± 0.03) compared with HFD, suggesting that MJM60668 could increase the lipid metabolism of HFD-treated mice (Figure 8c).

#### 3.5.6. MJM60668 Affects Fatty Acids Metabolism by Directly Affecting Adiponectin and Leptin Expression with an Anti-Inflammatory Effect on Liver

Adiponectin is a novel therapeutic target associated with diabetes and metabolic syndrome due to its anti-inflammatory and insulin-sensitizer effect [53]. Leptin is a key hormone for the regulation of energy and metabolism [54]. Leptin and adiponectin are released by the adipose tissue [54]. Our results showed that serum adiponectin levels in the HFD group were significantly decreased compared with those in the control group (575.76 ± 245.81 ng/mL vs. 848.34 ± 83.51 ng/mL). However, MJM60668 (10^8^) and MJM60668 (10^9^) significantly increased serum levels of adiponectin expression (947.37 ± 31.08 ng/mL and 855.04 ± 65.55 ng/mL, respectively) compared with the HFD group (Figure 5f).

As shown in Figure 5g, serum leptin levels were significantly increased in the HFD group (3895.49 ± 1515.92 pg/mL) compared with that in the control group (1469.18 ± 2597.2 pg/mL). However, MJM60668 (10^8^) and MJM60668 (10^9^) significantly decreased serum leptin levels (2964.61 ± 952.18 pg/mL and 2506.56 ± 567.67 pg/mL, respectively) increased by HFD.

Interleukin 1β (IL-1β) is a pro-inflammatory cytokine secreted by Kupffer cells. It has a key function in liver regeneration under inflammatory conditions [55]. Our results revealed that IL-1β serum levels were significantly increased in the HFD group (9.79 ± 0.6 pg/mL) than in the control group (8.7 ± 0.79 pg/mL). Administration of MJM60668 (10^8^) and MJM60668 (10^9^) significantly reduced serum levels of IL-1β to 8.14 ± 0.57 pg/mL and 8.43 ± 0.63 pg/mL, respectively, increased by HFD (Figure 5h). Taken together, these results suggest that MJM60668 can directly affect adipogenesis by decreasing serum leptin levels induced by HFD. It could also directly affect liver inflammation and regeneration by increasing serum levels of adiponectin and IL-1β in the body.

#### 3.5.7. Effects of MJM60668 on Intestinal Microbiota Diversity

The association between LAB intake and changes in gut microbiota was evaluated at the phylum and family levels. At the phylum level, as shown in Figure 9a, Firmicutes were predominant in the HFD group. Verrucomicrobia, a beneficial microbe in charge of intestinal health, was increased in MJM60668-treated mice. Portions of Actinobacteria, *Bacteroidetes, Deferribacteres*, *Firmicutes*, *Tenericutes*, *Verrucomicrobia*, and *Proteobacteria* were 0.8, 0.42, 0.02, 83.82, 0.1, 14.32, 0.1, and 0.4% in the control group, 1.1, 0.62, 0.17, 97.6, 0.3, 0.02, 0.07, and 0.1% in the HFD group, and 0.47, 0.7, 0.4, 95.7, 0, 2.65, and 0.05% in the MJM60668 (10^9^) group, respectively.

At the family level, mice microbiota was mainly composed of *Atopobiaceae*, *Bacteroidaceae*, *Deferribacteraceae*, *Chirstensenellaceae*, *Clostridiaceae* 1, Clostridiales vandinBB60 group, Family XIII, *Lachnospiraceae*, *Peptostreptococcaceae*, *Ruminococcaceae*, *Akkermansiaceaea*, and others (Figure 9b). Relative portions of *Atopobiaceae*, *Bacteroidaceae*, *Deferribacteraceae*, *Chirstensenellaceae*, *Clostridiaceae* 1, Clostridiales vandinBB60 group, Family XIII, *Lachnospiraceae*, *Peptostreptococcaceae*, *Ruminococcaceae*, *Akkermansiaceaea*, and others were 0.67, 0.17, 0.02, 0.02, 0.15, 1.8, 0.32, 48.55, 0, 32.8, 0.12, 14.32, and 1.02% in the control group, 1.07, 0.25, 0.17, 0.25, 0.27, 8.15, 0.15, 53.72, 0.6, 34.3, 0.32, 0, and 0.72% in the HFD group, and 0.47, 0.42, 0.4, 0.25, 0.17, 10.35, 0.1, 50.67, 0.35, 33.82, 0, 2.65, and 0.32% in the MJM60668 (10^9^) group. MJM60668 treated mice showed an increase in the Akkermansiaceae family known to be associated with a healthy gut barrier with an anti-inflammation effect on the gut and decrease weight gain in animals [56] (Figure 9b).

As shown in Figure 9d, at the genus levels, our result shows an increase in the Lachnospiraceae family on the silymarin and MJM60668 treated mice, compared to the HFD group. Additionally, increases in the Akkermansiaceae family are shown in the silymarin and MJM60668-treated mice, compared to the HFD group, in which we see a complete depletion of these families.

The Shannon Diversity index was evaluated to determine the gut microbial alpha diversity. The Shannon index was significantly lower in the HFD group that in the control group. There was a significative increase in the silymarin and MJM60668 treated mice compared to the HFD group (Figure 9e).

## 4. Discussion

The human body is the host of millions of microbes, with intestinal microbiota having the ability to directly influence the metabolism of the host [57]. A disruption of the balance of intestinal microbiota has a direct effect on host metabolism. It can contribute to the development of metabolic syndrome [57]. Previous studies have demonstrated the relation of gut microbiota imbalance with diabetes type 2 and obesity, both of which are strongly associated with the development of NALFD [58].

Probiotics are defined as “live microorganisms which, when administered in adequate amounts, confer a health benefit on the host” by the World Health Organization. The concept of administering probiotics as an alternative to antibiotics and safe treatment for various diseases to achieve health benefits has been increasing over the years due to the increase in antibiotic resistance [59]. The administration of probiotics can modulate intestinal microbiota and confer a beneficial effect on health [60]. Previous studies have shown that *Lactobacillus* can significantly change the compositions of the gut microbiota, increasing Bacteroidetes and Verrucomicrobia but decreasing Firmicutes in the host [61]. Studies on mice fed an HFD have shown that treatment with *Lactobacillus* and *Bifidobacterium* can change gut microbiota can increase insulin sensibility [62].

Among the *Lactobacillus* species, *Lactobacillus reuteri* is a well-studied probiotic bacterium that widely colonizes different parts of the human body, primarily the gastrointestinal tract by adhering to the intestinal epithelium. It can produce proteins that can bind the mucus, thereby restoring the balance of gut microbiota compositions of the host [63]. It can also enhance the immune system, reduce inflammation [64,65], reduce oxidative stress, and restore hepatic damage [66]. In a previous study, metabolomics analysis showed that *L. reuteri* attenuates alcoholic liver disease by interfering with fatty acid metabolic pathways. However, the effect of *L. reuteri* on NAFLD has not been investigated yet. In this study, *L. reuteri* MJM60668 was supplied to HFD-fed mice to evaluate the beneficial effect of *L. reuteri* on NAFLD.

In this study, supplementation of *L. reuteri* MJM60668 to HFD-induced NAFLD mice attenuated changed the gut microbiota population, especially increased the proportion of *Akkermansiaceae* in feces. *Akkermansiaceae* is a common bacteria found in the gut and responsible for degenerating mucin in the guts, its low presence in the gut microbiota leads to metabolic disorders such as diabetes [67]. This increase in MJM60668-treated mice can implicate an improvement in the gut barrier by increasing the mucin degeneration and decreasing the permeability of the gut, leading to improvement in the mucus barrier and decreasing the inflammation of the gut. Notably, the *Lachnospiraceae* NK4A136 group, which was significantly decreased in the HFD group, was recovered in the MJM60668-treated group. The role of the *Lachnospiraceae* NK4A136 group is not clear, but many reports showed that the NK4A136 group was reduced and negatively correlates to triglyceride in HFD mice. It was recognized as a potential probiotic recently [68]. However, the mechanism underlying this needs to be further investigated in the future.

Adipogenesis is regulated by a complex differentiation process induced by cytokines, hormones, and signaling pathways [69]. It has been shown that Lactobacillus can inhibit adipocyte differentiation [70]. In this study, treatment of MJM60668 (10^8^) or MJM60668 (10^9^) inhibited lipid accumulation in a dose-dependent manner in HepG2 cells (Figure 2). This indicated the potential of MJM60668 for the inhibition of adipogenesis. This activity was also confirmed in the animal study. *L. reuteri* MJM60668 suppressed weight gain, liver weight, and liver/body weight ratio in the high-fat mice model (Figure 4a,b). Because there is no significant difference in total caloric consumption, the inhibitory activity was caused by the administration of MJM60668.

In addition, many NAFLD symptoms were attenuated by the treatment of MJM60668. NAFLD is representative of the symptoms induced by obesity [71] characterized by the alteration of lipid metabolism in the liver, leading to a progressive lipid accumulation [72]. Liver biomarkers include body weight gain [43], steatosis [44], and injury (mainly indicated by ALT and AST) [41,42]. The result of our study showed that serum levels of AST and ALT in MJM60668-treated mice were significantly lower than those in the HFD group. This result suggests that *L. reuteri* MJM60668 can positively improve liver function by reducing lipid accumulation and promoting lipid metabolism in treated mice.

Lipidemia are one of the major complications that occur as a consequence of NAFLD [73] where obesity leads to TG accumulation in the adipose tissue for energy storage [69,70]. The significant increase in TG levels in the HFD group was consistent with an obesity characteristic. Administration of MJM60668 showed suppression of TGs levels in blood serum confirming the possibility of steatosis development prevention by daily administration of *L. reuteri* MJM60668 on obese mice. Cholesterol is an important molecule with a barrier function between cells and the environment for the formation of the raft for signaling molecules. Additionally, cholesterol plays a key role as a precursor for the synthesis of bile acids, vitamins, and hormones [74,75]. Excessive cholesterol will be retained in lipid droplets to decrease lipolysis and reduce fatty acids beta-oxidation in cells [76], Excessive cholesterol accumulation in the liver can lead to hepatic steatosis [77]. Excessive consumption of fat in the diet can increase cholesterol levels in the liver and increase high-density lipoprotein levels in humans and mice [78]. Our result showed a significant decrease of TCHO in MJM60668 treated mice than in the HFD group (Figure 5d).

NAFLD is characterized by increases of single and large intracytoplasmic lipids pushing aside hepatocellular nuclei and ballooned hepatocytes surrounded by a clear and vacuolar cytoplasm [45]. In the present study, ballooned hepatocytes were visualized and graded with a scale based on the percentage of balloon compared to the total number of hepatocytes in the liver sample, giving them scores based on the percentage of hepatocytes affected (score range: 0 to 3, where 0 = normal and 3 = severe damage) [47,48]. As shown in Figure 6, a significative reduction of steatosis grade was shown on MJM60668 treated mice in a dose-dependent manner, suggesting MJM606668 can be a potential probiotic strain to restore the liver damage that led to the steatosis.

Leptin is an adipocyte-derived hormone that regulates food intake and energy expenditure [79]. Due to leptin resistance, most NAFLD patients exhibit high leptin levels [80]. Our results showed that serum leptin levels in MJM60668-treated mice were markedly downregulated, suggesting leptin resistance was relieved.

At the same time, adiponectin was found to be elevated in the MJM60668-treated group (Figure 5f,g). Adiponectin is a key peptide hormone involved in obesity-related secretion in the adipocyte, and its concentration is low in NAFLD patients. Adiponectin binds to the receptor of hepatocyte, AdipoR, to regulate glucose and fatty acid metabolism via the activation of the AMPK and PPARα pathway. Therefore, in the next step, the downstream gene expression was investigated. These genes include genes related to fatty acid oxidation (PPARα, CPT1) or synthesis (ACC, FAS, SREBP).

Peroxisome proliferator-activated (PPAR)-alpha (PPARα) is the most abundant isotype in the liver and hepatocyte that regulate genes involved in fatty acid beta-oxidation and homeostasis in the body regulating the expression of genes and metabolic function expressing its receptor into adipose tissue [52] Its deletion may lead to NAFLD promotion [81] otherwise, an increase of PPAR activity can improve HFD-induced dysbiosis and weight gain in mice [82]. CPT1A is an active gene in the mitochondria of liver cells [83], It is a rate-limiting enzyme in fatty acid oxidation. It regulates the rate of fatty acid synthesis [84]. In the liver, CPT1A gene expression can be induced by starvation, HFD, and PPARα ligand [52]. In this study, an upregulation of PPARα and CPT1A on MJM60668 treated mice suggests that our strain administration can upregulate the fatty acid oxidation and reduce the lipid accumulation that contributes to weight loss. Acetyl-CoA carboxylase (ACC) is an essential substrate needed for fatty acid synthesis in lipogenic tissues [85]. FAS is a death receptor involved in apoptosis developed from liver steatosis in humans [86]. Our results confirmed that the oral administration of MJM60668 to HFD-fed mice could significantly decrease FAS and ACC expression (Figure 7a,b and Figure 8a) and reduce fatty acids oxidation. FAS and ACC are positively regulated by SREBP transcription in lipogenesis. They are positively related to NALFD development in patients by impairing the regulation of fatty acids synthesis related to SREBP upregulation [87]. Our results showed that administration of MJM60668 could positively downregulate the relative expression of SREBP in the liver and reduce glycolytic and lipogenic enzymes such as FAS and ACC, confirming its ability to inhibit lipogenesis and glycolysis in treated mice [88].

Previous report has shown that NALFD develop pathway is responsible for activation of IL-1 β, which contributes to liver disease by activating pro-inflammatory cytokines [89]. Our result showed a downregulation of IL-1 β in MJM60668 treated mice, showing a reduction of pro-inflammatory cytokines activation in treated mice compared with HFD group. Additionally, levels of IL-6, responsible for the induction of hepatic acute phase response [90] have shown a reduction in MJM60668 treated mice, suggesting our strain administration can reduce inflammation developed by the increase of fat consumption in HFD feed mice.

## 5. Conclusions

In conclusion, *L. reuteri* MJM60668 can positively improve gut microbiota dysbiosis, markedly inhibit the inflammation characteristic of non-alcoholic fatty liver disease, and significantly inhibit adipogenesis by regulating serum adiponectin levels in treated mice.

## Figures and Tables

**Figure 1 microorganisms-10-02203-f001:**
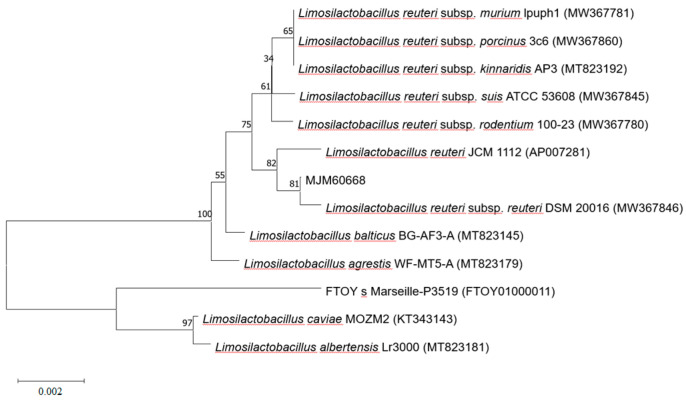
Phylogenic analysis of MJM60668. Phylogenetic analysis was performed by the neighbor-joining method, using MEGA11 software. Numbers on the branches represent the bootstrap values (%) from 1000 replicates. The evolutionary distances were computed using the Kimura 2-parameter method. The scale bar indicates substitutions of 0.002 per nucleotide position.

**Figure 2 microorganisms-10-02203-f002:**
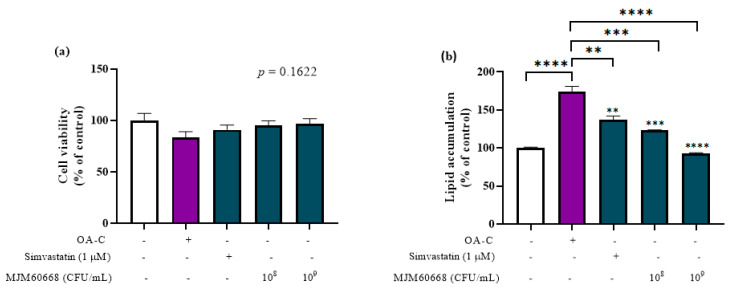
Effect of MJM60668 on HepG2 cells. HepG2 cells were stimulated with 1 nM Oleic acid and 7.5 µg/mL Cholesterol (OA-C) and treated with MJM60668 for 6 h. Simvastatin (1 µM) was used as a positive control. (**a**) Cell viability was measured by MTT assay. (**b**) Lipid accumulation on HepG2 cells was determined by the Oil Red O staining method. Results are presented as the mean ± standard deviation of triplicate independent experiments. ** *p* < 0.01, *** *p* < 0.001, **** *p* < 0.0001 compared with the OA-C treatment.

**Figure 3 microorganisms-10-02203-f003:**
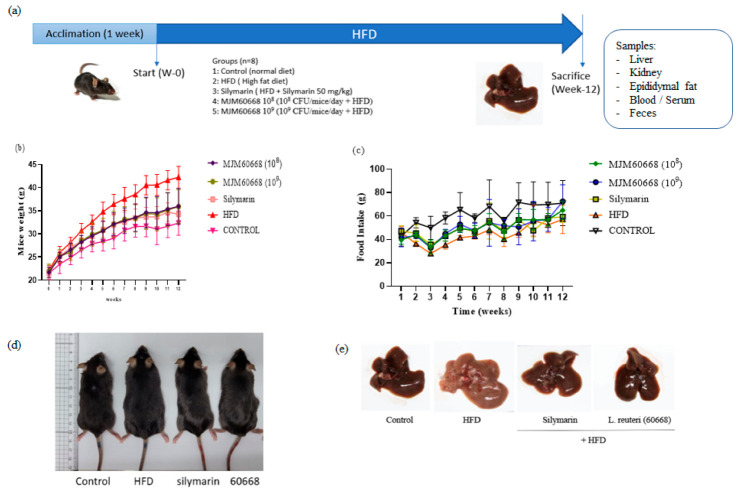
(**a**) Diagram of mouse model. Mice were divided into 5 groups: Control group, HFD group, Silymarin group, MJM60668 (10^8^) group, and MJM60668 (10^9^) group. The HFD, Silymarin, and MJM60668 groups were fed an HFD diet and free water daily. Moreover, mice in the Silymarin group were also treated with silymarin at (50 mg/kg), MJM60668 (10^8^) were treated with *L. reuteri* at a concentration of 1 × 10^8^ CFU, daily, and MJM60668 (10^9^) were treated with *L. reuteri* at a concentration of 1 × 10^9^ CFU, daily. (**b**) body weight recording weekly for 12 weeks of the experiment. (**c**) Food intake per cage recording weekly. (**d**) Comparative image with body shape mice before sacrifice. (**e**) Image with liver after sacrifice.

**Figure 4 microorganisms-10-02203-f004:**
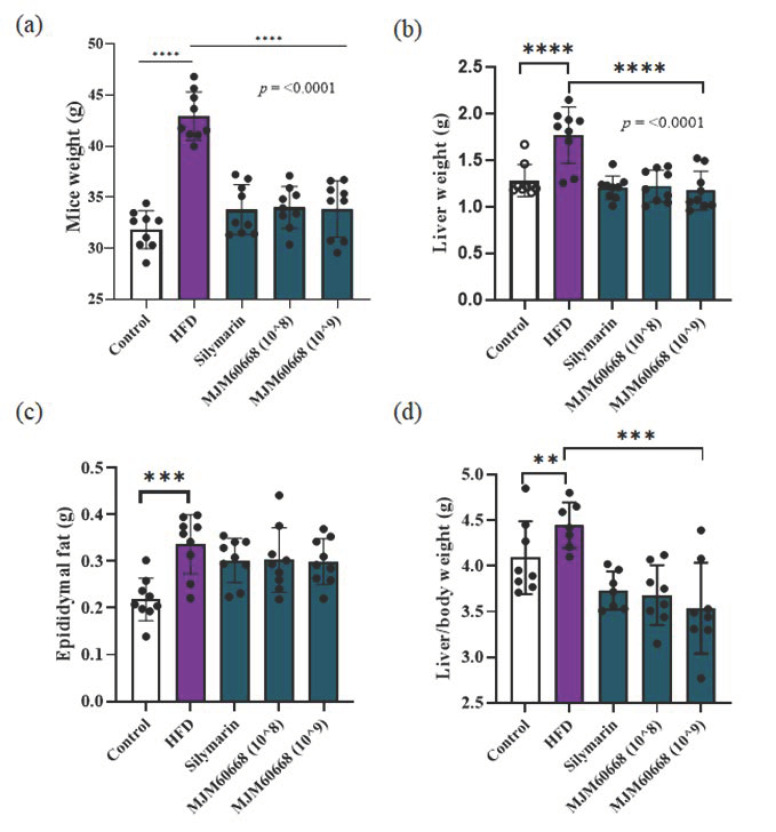
*L. reuteri* MJM60668 treatment reduces body weight, liver weight, epididymal weight, and liver/body ratio in mice. (**a**) Mice weight after sacrifice (**b**) Liver weight. (**c**) Epididymal fat, (**d**) Liver to body weight ratio represented as the mean ± standard of result. ** *p* < 0.01, *** *p* < 0.001, **** *p* < 0.0001 compared with HFD group.

**Figure 5 microorganisms-10-02203-f005:**
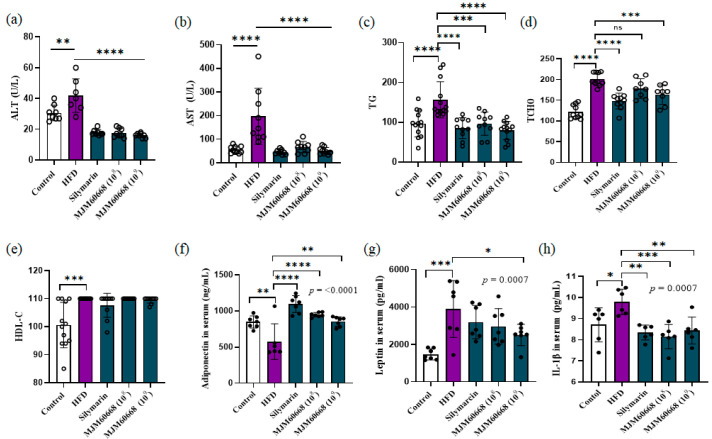
*L. reuteri* MJM60668 reduces hepatic lipid accumulation and liver injury in mice. (**a**–**e**) Biochemical analysis of serum samples, ALT: Alanine Aminotransferase, AST: Aspartate Aminotransferase, TG: Triglyceride, TCHO: Total Cholesterol, HDL-C: High-density lipoprotein cholesterol. (**f**–**h**) MJM60668 increases leptin and decreases adiponectin levels and decreases IL-1β. (**f**) Leptin level on serum. (**g**) Adiponectin level on serum. (**h**) IL-1β level on serum. Data represented as mean ± S.E.M of three independent experiments. *n* = 12. ns = non-significant differences, * *p* < 0.05, ** *p* < 0.01, *** *p* < 0.001, **** *p* < 0.0001 compared with HFD group.

**Figure 6 microorganisms-10-02203-f006:**
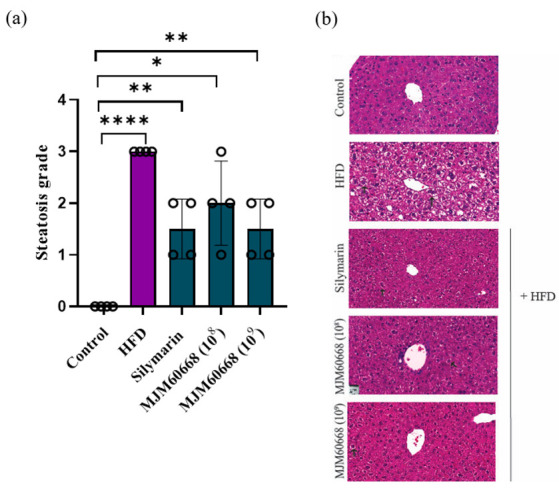
Liver biopsy of Non-alcoholic fatty liver disease (NAFLD) and steatosis grade. (**a**) Steatosis grade of the liver based on ballooned hepatocyte: 0 (normal, <5%), 1 (mild, 5–33%), 2 (moderate, 34–66%), 3 (severe, >66%). (**b**) Histologic features of NAFLD include ballooning and lobular inflammation. Arrow indicates ballooning hepatocyte. Image at a magnification of 20× H&E staining. Data represented as mean ± S.E.M of three independent experiments. *n* = 4. * *p* < 0.05, ** *p* < 0.01, **** *p* < 0.0001 compared with control group.

**Figure 7 microorganisms-10-02203-f007:**
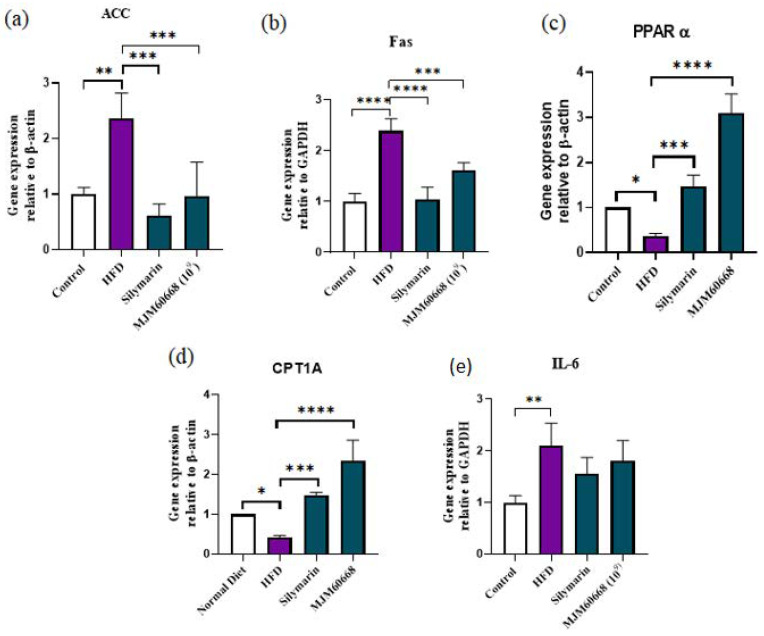
The effect of *L. reuteri* MJM60668 on mRNA expression in hyperlipidemia mouse (**a**) the mRNA expression of ACC in mice liver, (**b**) mRNA expression of Fas in mice liver, (**c**) mRNA expression of PPARα in mice liver, (**d**) mRNA expression of CPT1A in mice liver, (**e**) mRNA expression of IL-6 in mice liver. Data represented as mean ± S.E.M of three independent experiments. *n* = 4. * *p* < 0.05, ** *p* < 0.01, *** *p* < 0.001, **** *p* < 0.0001 compared with HFD group.

**Figure 8 microorganisms-10-02203-f008:**
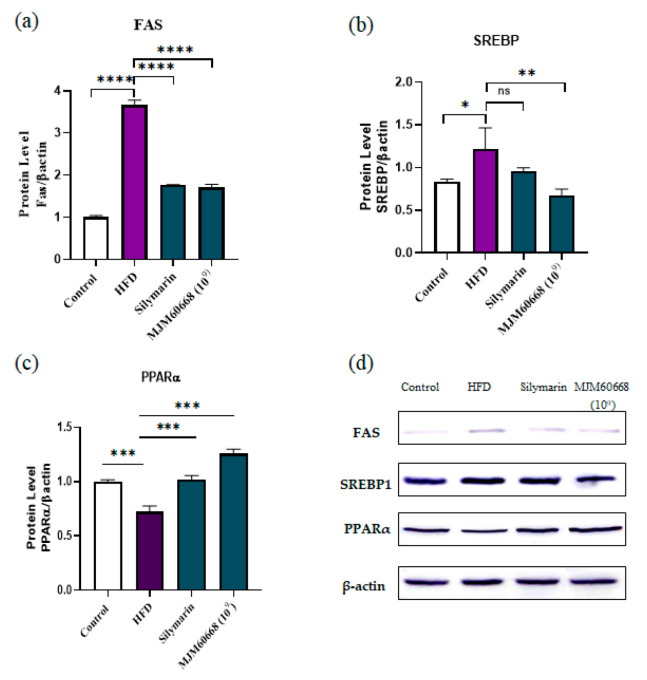
*L reuteri* MJM60668 decreases FAS and SREBP and increases PPARα expression in liver at protein level. (**a**) protein expression of FAS in mice liver, (**b**) protein expression of SREBP in mice liver. (**c**) protein expression of PPARα in mice liver. (**d**) Images for western blot band. Data represented as mean ± S.E.M of three independent experiments. *n* = 4. ns = non-significant differences * *p* < 0.05 compared with HFD group, ** *p* < 0.01, *** *p* < 0.001, **** *p* < 0.0001 compared with HFD group.

**Figure 9 microorganisms-10-02203-f009:**
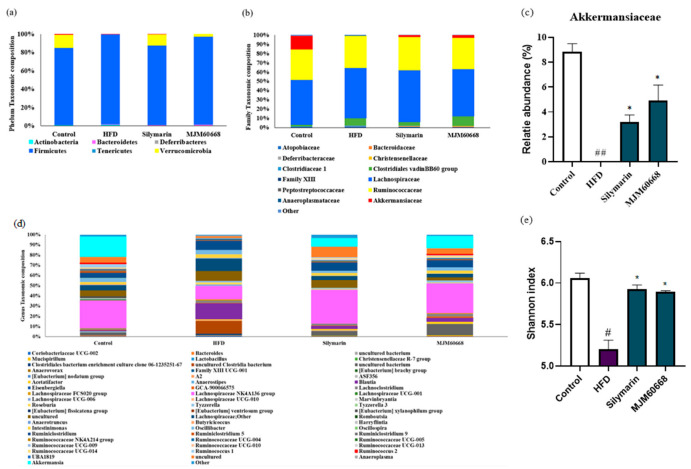
LAB supplementations modulate the composition of gut microbiota. (**a**) taxonomic composition at the phylum level. (**b**) Taxonomic composition at the family level (**c**) relative abundance of Akkermansiaceae. (**d**) taxonomic composition at the genus level. (**e**) Alpha-diversity is indicated by the Shannon index. ^#^
*p* < 0.05, ^##^
*p* < 0.01 compared with control group, * *p* < 0.05 compared with HFD group.

**Table 1 microorganisms-10-02203-t001:** Sequences of primers (mouse) used in RT-qPCR.

Name	Primer	Sequence (5′ to 3′)
FAS	Forward	AGGGGTCGACCTGGTCCTCA
	Reverse	GCCATGCCCAGAGGGTGGTT
ACC	Forward	AACATCCCGCACCTTCTTCTAC
	Reverse	CTTCCACAAACCAGCGTCTC
PPARα	Forward	AGAGCCCCATCTGTCCTCTC
	Reverse	ACTGGTAGTCTGCAAAACCAAA
CPT1A	Forward	TGGCATCATCACTGGTGTGTT
	Reverse	GTCTAGGGTCCGATTGATCTTTG
IL-6	Forward	ACAACCACGGCCTTCCCTACTT
	Reverse	CACGATTTCCCAGAGAACATGTG
β-actin	Forward	ACAACCACGGCCTTCCCTACTT
	Reverse	CACGATTTCCCAGAGAACATGTG

**Table 2 microorganisms-10-02203-t002:** Safety assessment of *L. reuteri* MJM60668.

Safety Test	*L. reuteri* MJM60668 (MJM60668)	LGG
Antibiotics *		
Ampicilin	1	1
Vancomycin	512 (NR)	512 (NR)
Gentamicin	8 (R)	32 (R)
Kanamycin	R	R
Streptomycin	128 (R)	32 (R)
Tetracycline	16	1
Clindamycin	1	1
Erythromycin	1	1
Chloramphenicol	4	4
D-lactate production	-	-
Bile salt deconjugation	-	-
Bioamin production	-	-
L-Histidine	-	-
L-Tyrosine	-	-
L-phenylalanine	-	-
Arginine	-	-
Tryptophan	-	-
L-ornithine	-	-
Mucin degradation	-	-
Hemolytic activity Adhesion activity	- 5.1 ± 0.43%	- 3.1 ± 0.29%

NR not required, R resistant, - no activity * MIC value for the antibiotics recommended by European food safety authority (EFSA), 2012.

**Table 3 microorganisms-10-02203-t003:** Antibacterial activity of *L. reuteri* MJM60668 and *L. rhamnosus* GG.

Strains	Diameter of Zone Inhibition (mm)	
	*L.ruteri* MJM60668	LGG
*Salmonella gallirarum* KCTC 2931	10	10
*Escherichia coli* K99	8	8
*Escherichia coli* O1 KCTC 2441	10	8
*Escherichia coli* 0138	6	6
*Escherichia coli* ATCC25922	8	8
*Salmonella chloreraesuis* KCTC 2932	10	8
*Salmonella typhi* KCTC 2514	8	8
*Pseudomonas aeruginosa* KCCM 11802	10	10

**Table 4 microorganisms-10-02203-t004:** Antibacterial activity of *L. reuteri* MJM60668 and *L. rhamnosus* GG.

OGI Transit		(Log_10_ CFU/mL)	
		MJM60668	LGG
Initinal		9.20 ± 0.048	9.12 ± 0.165
Oral stress	-	9.19 ± 0.04	9.09 ± 0.039
	+	9.09 ± 0.1	9.00 ± 0.006
Gastric stress (pH3)	-	9.16 ± 0.06	9.07 ± 0.026
	+	9.06 ± 0.042	8.84 ± 0.052
Gastric stress (pH2)	-	9.18 ± 0.053	9.05 ± 0.007
	+	8.97 ± 0.027	8.74 ± 0.061
Intestinal stress	-	9.15 ± 0.009	9.08 ± 0.092
	+	8.68± 0.006	7.79 ± 0.095

- without stress, + with stress.

## Data Availability

The data support the findings of this study are available from the corresponding author upon reasonable request.

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
