# Peer review of "Lactobacillus reuteri MJM60668 Prevent Progression of Non-Alcoholic Fatty Liver Disease through Anti-Adipogenesis and Anti-Inflammatory Pathway"

_microorganisms, 2022, doi:10.3390/microorganisms10112203_

Round 1

Reviewer 1 Report

The author describes the effect of a strain of Lactobacillus reuteri on NAFLD. Although the effect is significant, there are still some aspects that need to be improved.

1. The author's description on line 71 that the effect of L. reuteri on NAFLD has not been reported is incorrect, in fact there have been some studies on L. reuteri on NAFLD, e.g. "PMID: 34776827, 26516851, 33977963"

2. It is suggested to further confirm whether the Lactobacillus reuteri in the text is a new strain through sequence alignment

3. Liver Oil Red O staining can better reflect the accumulation of lipid droplets

4. The data display of the flora analysis in Figure 9 is not comprehensive enough, it does not show the changes of flora diversity, and the analysis at the genus level

Author Response

Dear reviewers.

Firstly, thank you for carefully reading our work and give us this manuscript review. Your comments help us to improve the quality of the draft. We revised the contents accordingly to your comments, hopping our revision help to answer your questions about this work. The point-to-point response to the reviewer’s comments are given below. 

Reviewer 1:

Review comments of manuscript microorganisms-1932694 entitled:

Lactobacillus reuteri MJM60668 prevent progression of non-alcoholic fatty liver disease through anti-adipogenesis and anti-inflammatory pathway

The author describes the effect of a strain of Lactobacillus reuteri on NAFLD. Although the effect is significant, there are still some aspects that need to be improved.:

  1. The author's description on line 71 that the effect of L. reuteri on NAFLD has not been reported is incorrect, in fact there have been some studies on L. reuteri on NAFLD, e.g. "PMID: 34776827, 26516851, 33977963"
  • Firstly, thank you very much for your comments on this draft. According to your indications, we have modified and correct the introduction by adding suggested and new references to this work.
  • (lines 68 to 76) “A previous study revealed lactobacillus reuteri can reduce fat accumulation, steatosis [22], and fibrosis [23] in high fat diet treated mice. Also L. reuteri strain has shown a positive effect in LPS-induced intestinal tight junction protein destruction and decreasing intestinal and inhibit intestinal and hepatic inflammatory signals in piglets. [24] Additionally has shown an effect as positive regulator of gut microbiota and improve metabolic system by increasing short-chain fatty acids (SCFAs) [25].

  1. It is suggested to further confirm whether the Lactobacillus reuteri in the text is a new strain through sequence alignment
  • MJM60668 was a new strain which was originally isolated in our lab from a fecal sample of a baby. Based on the 16S rRNA full sequence, this strain MJM60668 showed 99.73% similarity to a standard strain Lactobacillus reuteri DSM 20016. So this strain was designated Lactobacillus reuteri MJM60668 in our study.
  1. Liver Oil Red O staining can better reflect the accumulation of lipid droplets
  • We agreed with your observation regarding an Oil Red O staining can better reflect the accumulation of lipid in droplets. Nevertheless, references support the analysis of steatosis can be performed using the data provided by the H&E staining. Since H&E is an easy and low cost approach to identify the lipid accumulation in different tissue and is clinically approved method to score steatosis. We selected this method only because we don’t have the equipment to perform Oil Red O staining in our laboratory.

References:

             doi:10.1002/hep.20701

doi:10.1016/j.metabol.2015.11.008

doi:10.1016/j.ejphar.2020.173461

  1. The data display of the flora analysis in Figure 9 is not comprehensive enough, it does not show the changes of flora diversity, and the analysis at the genus level
  • As your suggestion, we added extra data on the figure 9 to complete the information related with the flora diversity and analysis at the genus level to make this information more comprehensive.
  • (lines 600 to 608): “As shown in the Figure 9d, at the genus levels, the Lachnospiraceae species was increased in the Silymarin and MJM60668 treated mice, compared to the HFD group. Additionally, increase on the Akkermansiacea species was shown in the Silymarin and MJM60668 treated mice, compared to the HFD group, in which showed a complete depletion of these family.

The Shannon Diversity index was evaluated to determine the gut microbial alpha diversity. Shannon index was significantly lower in the HFD group that in the control group. There was a significative increase in the Sylimarin and MJM60668 treated mice compared to the HFD group (Figure 9e)”

Reviewer 2 Report

In this paper, the authors showed MJM60668 alleviated HFD-induced fatty liver disease. I have a few questions about the results.

1) Is there any clinical data on the relationship between the amount of L. reuteri MJM60668 and fatty liver disease? For example, do patients with fatty liver disease have high or lower amount of L. reuteri? If no such data exists, in your HFD-induced fatty liver disease model, do mice fed with HDF have higher or lower amount of L. reuteri compared to control group?

2) Is there a way to deplete L. reuteri in mouse model. If so, can you deplete L. reuteri in mice and then feed them with HFD to see if they will develop more severe disease phenotype?

3) Is there any difference in the immune cell profile? Since fatty liver disease is an inflammatory disease, changes in immune responses could possibly affect the disease progression, and result in the phenotypes you saw. 

4) Does L. reuteri protect against fatty liver disease or promote the recovery from the disease. Can you try to give mice L. reuteri after they have developed liver disease to see if L. reuteri still has any beneficial effects. 

Author Response

Dear reviewers.

Firstly, thank you for carefully reading our work and give us this manuscript review. Your comments help us to improve the quality of the draft. We revised the contents accordingly to your comments, hopping our revision help to answer your questions about this work. The point-to-point response to the reviewer’s comments are given below. 

Reviewer 2:

Review comments of manuscript microorganisms-1932694 entitled:

Lactobacillus reuteri MJM60668 prevent progression of non-alcoholic fatty liver disease through anti-adipogenesis and anti-inflammatory pathway

In this paper, the authors showed MJM60668 alleviated HFD-induced fatty liver disease. I have a few questions about the results:

  1. Is there any clinical data on the relationship between the amount of L. reuteri MJM60668 and fatty liver disease? For example, do patients with fatty liver disease have high or lower amount of L. reuteri? If no such data exists, in your HFD-induced fatty liver disease model, do mice fed with HDF have higher or lower amount of L. reuteri compared to control group?"
  • Until now, there is no report on the relation between reuteri and NAFLD. However, many research showed the potential of a probiotic on the treatment of NAFLD/NASH. In our study, we chose L. reuteri MJM60668 strain for animal study because this strain showed anti-lipid accumulation effect on HepG2 cells in the cell screening. Besides, in microbiota analysis, the total Lactobacillus genus was detected less than 0.1% of the whole intestinal microbiota. It is very hard to figure out the proportion of a single species based on the 16S rRNA based metagenome analysis. This is a shortage of this method. However, treatment with MJM60668 significantly attenuated NAFLD in mouse model, and changed the composition of gut microbiota. We therefore speculated the ameliorative effect of L. reuteri MJM6668 is a comprehensive result of its modulation on the gut microbiota, not only the activity exerted by the strain directly.
  1. Is there a way to deplete L. reuteri in mouse model. If so, can you deplete L. reuteri in mice and then feed them with HFD to see if they will develop more severe disease phenotype?
  • Firstly, thank you for your comment. Mice used in this study are SPF, for this reason the Lactobacillus is unavoidably presented in the mice. Our mice facility doesn’t have the equipment to work with germ-free and place a complete depletion of reuteri in which doesn’t have any route to transfer any Lactobacillus to treated mice.
  • With the progress of this study, if we can find some evidence that depletion of L.reuteri is related to NAFLD/NASH, we will try to test if the reuteri depletion will severe the NAFLD caused by HFD in a germ-free mice according to your suggestion.  

  1. Is there any difference in the immune cell profile? Since fatty liver disease is an inflammatory disease, changes in immune responses could possibly affect the disease progression, and result in the phenotypes you saw.
  • Based on our result showed in the Figure 7, we focus to identify the effect of L. reuteri MJM60668 on the immune response by the identification of IL-6 levels expressed in hepatic tissue, since IL-6 is a cytokine with broad-ranging effect with integrated immune response and IL-6 levels can show a immunocompetence of the body against pathogens.

Our data shown IL-6 has no significative difference with HFD treated mice (P = 0.7767) suggesting the immune profile slightly change into a less active, for this reason we don’t proceed with analyze most deeply immune response effect in this work because was not the principal objective of our hypothesis. Nevertheless, we will consider stablish the option of check the complete immune cell profile in our future work in which L. reuteri was administered after induction of NALFD in mice.   

References:

doi: 10.1111/j.1572-0241.2007.01774.x.

https://doi.org/10.1093/qjmed/hcaa052.048

https://doi.org/10.3389/fimmu.2021.708959

  1. Does L. reuteri protect against fatty liver disease or promote the recovery from the disease. Can you try to give mice L. reuteri after they have developed liver disease to see if L. reuteri still has any beneficial effects
  • Our first experiment was carried out based on the hypothesis that reuteri can have a preventive effect on the development and progression of fatty liver disease, for this reason in this work we carried out the treatment at the same time with NALFD induction in mice and treatment model was not the objective of this work. Nevertheless, based on our currently result, our future work will consider corroborate the effectiveness of L. reuteri as a treatment of NALFD, where we will start the treatment with L. reuteri in mice after inducing liver disease.

Round 2

Reviewer 1 Report

The authors had revised the suggestion that I made.